# UNPAIRED PREFERENCE OPTIMIZATION: ALIGNING VISUAL GENERATIVE MODELS WITH SCALAR FEEDBACK

## ABSTRACT

Direct Preference Optimization (DPO) provides a stable and simple alternative to reinforcement learning for aligning large generative models, yet its dependence on paired comparisons remains a critical limitation. In practice, feedback is often collected as unpaired scalar scores, such as human ratings, which cannot be directly used by DPO. To resolve this, we first revisit the KL-regularized alignment objective and show that for individual samples, the optimal policy is determined by an elegant but intractable decision rule: comparing a sample's reward against an instance-dependent oracle baseline. Building on this insight, we introduce Unpaired Preference Optimization (UPO), a new framework that provides a principled and tractable proxy for this ideal rule. UPO approximates this oracle baseline with a global threshold derived from empirical score distribution, thereby reframing alignment as a classification task on unpaired data. This core mechanism is further enhanced by a confidence-weighting scheme to leverage the full magnitude of the scores. Extensive experiments demonstrate that UPO effectively aligns diverse generative models, including both diffusion and MaskGIT paradigms, significantly outperforming standard fine-tuning baselines. By extending the simplicity of DPO to the more practical setting of unpaired scalar feedback, UPO provides a principled and scalable path for aligning generative models with human preference signals.

## 1 INTRODUCTION

Aligning large generative models with human preference has become a central challenge in post-training. Reinforcement Learning from Human Feedback (RLHF) (Christiano et al., 2017; Achiam et al., 2023; Ouyang et al., 2022; Stiennon et al., 2020) was a seminal paradigm, demonstrating that models can be optimized to align complex human values. But its operational complexity and training instability (Rafailov et al., 2023) have motivated a shift towards simpler and more stable "policy fitting" methods, Direct Preference Optimization (DPO). DPO reframes the alignment problem as a simple classification loss over pairs of preferred ($y_w$) and rejected ($y_l$) responses. Following prior work (Peters & Schaal, 2007; Peng et al., 2019; Korbak et al., 2022; Go et al., 2023), by leveraging a closed-form solution to the KL-regularized reinforcement learning objective, DPO bypasses the need for explicit reward modeling and the instabilities of RL training. This approach is effective because the intractable partition function, $Z(x)$, cancels out when computing the difference between two responses (Rafailov et al., 2023), and is equivalent to fitting a reparametrized Bradley-Terry model (Bradley & Terry, 1952).

However, the primary limitation of DPO (Rafailov et al., 2023) and its successors (Meng et al., 2024; Ethayarajh et al., 2024; Liu et al., 2024) is their fundamental reliance on paired preference data (preferred vs. dis-preferred). In many practical scenarios, feedback is more naturally collected as unpaired samples with absolute scores. For example, -5 to 5 star ratings (*i.e.*, peer-review ratings) from users or scalar outputs from a reward model. Such unpaired data cannot be easily transformed into preference pairs. This results in a fundamental gap between DPO's reliance on paired preference data and the unstructured nature of many real-world preference signals.

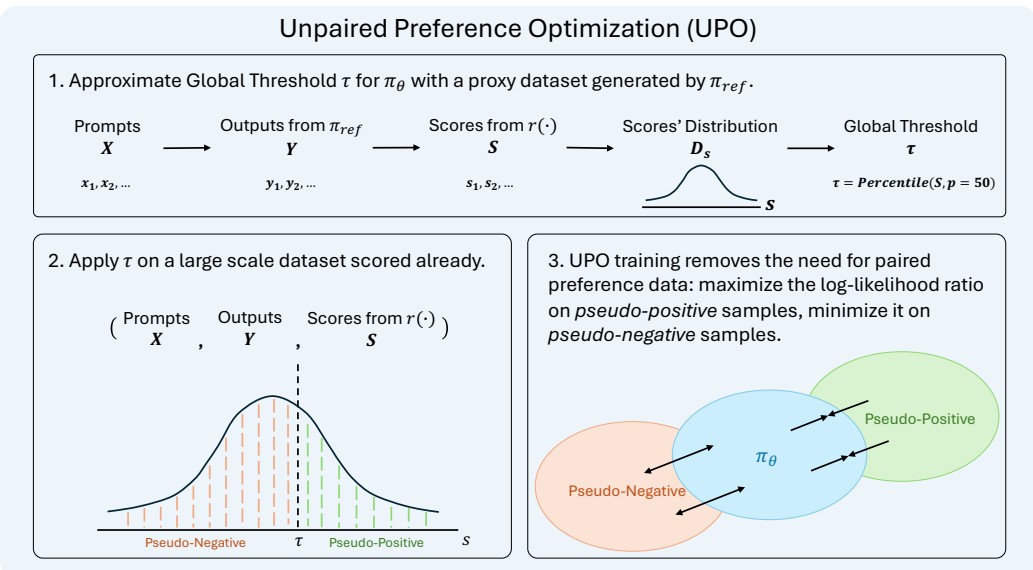

Figure 1: The overview of unpaired preference optimization.

To bridge the gap, we introduce **Unpaired Preference Optimization (UPO)**, a novel alignment algorithm that extends the stability and simplicity of the DPO framework to operate on unpaired, scored data, as presented in Figure 1. The core insight of UPO is to dynamically generate preference signals from absolute scores through a **thresholding mechanism**. By classifying samples as "pseudo-preferred" or "pseudo-rejected" relative to a global threshold (*e.g.*, the median preference score of current policy), UPO constructs the necessary signal for a preference-style loss without requiring explicit pairs. Furthermore, UPO introduces a **reward-weighting mechanism** that modulates the loss for each sample based on the magnitude of its score relative to the threshold. This allows the model to learn more from high-confidence examples, fully leveraging the information latent in absolute scores.

Taking a more theoretical approach, we revisit the KL-regularized alignment objective (Eq. 1) and demonstrate that for any individual sample, the optimal policy's decision on whether to increase or decrease its probability relative to a reference model is governed by an elegant but intractable decision rule: the sample's reward must be compared against an instance-dependent baseline: $\tau^*(x) = \beta \log Z(x)$. This reveals the core theoretical challenge that has previously limited direct methods to both paired and unpaired data: the intractability of the oracle baseline $\tau^*(x)$. UPO resolves this challenge by introducing a **principled and tractable proxy for this ideal rule**. Specifically, UPO approximates the instance-dependent oracle baseline with a global threshold $\tau$ derived from empirical score distribution. This reframes the alignment problem as a binary classification on unpaired data, where samples are labeled as "pseudo-preferred" or "pseudo-rejected". This core mechanism is further enhanced by a confidence-weighting scheme that leverages the full informational richness of the scalar scores.

We validate UPO under two dominant paradigms in vision-centric generative modeling: diffusion-based training (Ho et al., 2020) using mean squared error (MSE) loss, and MaskGIT-style masked token modeling (Chang et al., 2022) using cross-entropy loss. Our empirical evaluation covers two popular foundation models (Stable Diffusion v1.4 (Rombach et al., 2022) and Meissonic (Bai et al., 2024) ) and four widely used reward models (HPSv2.1 (Wu et al., 2023), PickScore (Kirstain et al., 2023), ImageReward (Xu et al., 2023), and LAION Aesthetic Score (Schuhmann et al., 2022)). Results consistently show that UPO aligns generative outputs with human preferences more effectively than direct supervised fine-tuning. Extensive ablation studies further provide guidance on hyperparameter selection and reveal bottlenecks in the online UPO setting. Together, these findings position UPO as a unified and scalable framework for preference alignment beyond the limitations of pairwise supervision. In summary, our contributions are as follows:

- We propose Unpaired Preference Optimization (UPO) for optimizing vision generative models, a simple yet effective method to align models using unpaired, scored data, addressing a key limitation in current policy fitting algorithms.

- We provide a principled derivation for UPO, framing it not as a heuristic but as a tractable approximation of the ideal KL-optimal decision rule. This insight leads to our novel global thresholding and weighting mechanisms.
- We demonstrate through comprehensive experiments on different generative paradigms that UPO successfully aligns generative models with various reward models, outperforming supervised fine-tuning baselines.

## 2 METHOD

In this section, we first review the KL-regularized RL objective that underpins modern alignment methods. We then analyze how existing policy fitting methods like DPO require paired data to ensure tractability. Finally, we formalize how UPO overcomes this constraint through its proxy objective, novel thresholding and weighting mechanisms, enabling direct learning from unpaired, absolute scores.

### 2.1 PRELIMINARIES: KL-REGULARIZED RL AND POLICY FITTING

The objective for many alignment methods is to find a policy $\pi_\theta$ that maximizes the expected reward while remaining close to a reference policy $\pi_{ref}$ (Ziebart et al., 2010; Jaques et al., 2019). This is formally expressed as:

$$\max_{\pi_\theta} \mathbb{E}_{x\sim\mathcal{D}, y\sim\pi_\theta(\cdot|x)}[\mathcal{R}(x,y)] - \beta\mathbb{D}_{KL}(\pi_\theta(\cdot|x)||\pi_{ref}(\cdot|x)) \tag{1}$$

where $\mathcal{R}(x,y)$ is a reward function that scores the quality of a completion $y$ for a prompt $x$, and $\beta$ is a hyperparameter controlling the strength of the KL-divergence penalty.

This objective has a closed-form optimal solution given by:

$$\pi^*(y|x) = \frac{1}{Z(x)}\pi_{ref}(y|x)\exp\left(\frac{1}{\beta}\mathcal{R}(x,y)\right) \tag{2}$$

where $Z(x) = \sum_y \pi_{ref}(y|x)\exp(\frac{1}{\beta}\mathcal{R}(x,y))$ is the per-prompt partition function. Direct optimization via this solution is generally intractable because computing $Z(x)$ requires summing over all possible completions $y$, an infinite space for language and vision models.

To avoid computing $Z(x)$, DPO (Rafailov et al., 2023) reparameterizes the reward function by expressing Eq. 2 in logarithmic form. Taking the logarithm of both sides and rearranging, we obtain:

$$\mathcal{R}(x,y) = \beta\log\frac{\pi_\theta(y|x)}{\pi_{\text{ref}}(y|x)} + \beta\log Z(x), \tag{3}$$

and observes that in pairwise preference models such as Bradley–Terry (Bradley & Terry, 1952), only the difference of rewards matters:

$$p(y_w \succ y_l|x) = \sigma\left(\mathcal{R}(x,y_w) - \mathcal{R}(x,y_l)\right), \tag{4}$$

where $\sigma(\cdot)$ denotes the logistic function. Plugging Eq. 3 into Eq. 4, the partition term $\log Z(x)$ cancels:

$$\mathcal{R}(x,y_w) - \mathcal{R}(x,y_l) = \beta\log\frac{\pi_\theta(y_w|x)}{\pi_{\text{ref}}(y_w|x)} - \beta\log\frac{\pi_\theta(y_l|x)}{\pi_{\text{ref}}(y_l|x)}. \tag{5}$$

This leads to the DPO loss:

$$\mathcal{L}_{\text{DPO}}(\pi_\theta;\pi_{\text{ref}}) = -\mathbb{E}_{(x,y_w,y_l)\sim\mathcal{D}}\left[\log\sigma\left(\beta\log\frac{\pi_\theta(y_w|x)}{\pi_{\text{ref}}(y_w|x)} - \beta\log\frac{\pi_\theta(y_l|x)}{\pi_{\text{ref}}(y_l|x)}\right)\right]. \tag{6}$$

Thus, DPO avoids estimating the intractable partition function $Z(x)$ altogether, enabling policy learning using only a classification-style objective.

While effective, this formulation is fundamentally tied to the availability of paired preference data $(y_w, y_l)$. That is, each training example must provide both a relatively better and a worse sample, annotated with respect to the same input $x$. This reliance on relative signals is both the strength and the primary limitation of existing policy fitting frameworks: they avoid explicit reward modeling, but cannot directly handle *unpaired* or *absolute* preference signals that are common in real-world data collection settings. Addressing this gap requires extending policy learning objectives beyond pairwise comparisons to more flexible forms of supervision.

## 2.2 Unpaired Preference Optimization (UPO)

While methods like DPO have proven effective, they are fundamentally constrained to datasets with explicit pairwise preferences $(y_w, y_l)$. In many real-world scenarios, supervision is available in a more granular, yet unpaired, format: a dataset $\mathcal{D} = \{(x_i, y_i, s_i)\}$ of samples with absolute scalar scores, such as human ratings or reward model outputs. Our goal is to leverage this prevalent data format to optimize the same KL-regularized objective in Eq. 1.

Directly applying the preference-based paradigm to unpaired data is challenging. The key insight of DPO, by canceling the intractable partition function $Z(x)$ through a log-ratio of preferences, is contingent on the availability of pairs. With only individual scored samples, this cancellation is no longer possible. UPO addresses this challenge by formulating a **principled proxy objective** that guides the policy $\pi_\theta$ towards higher-reward regions without requiring explicit computation of $Z(x)$ or access to preference pairs.

### 2.2.1 From Pairwise Cancellation to a Pointwise Intractability

To understand the core challenge, we revisit the optimal policy $\pi^*$ from Eq. 2. By taking the logarithm and rearranging, we can express the relationship between the optimal and reference policies for a single sample $(x, y)$:

$$\log \frac{\pi^*(y|x)}{\pi_{\text{ref}}(y|x)} = \frac{1}{\beta}\mathcal{R}(x, y) - \log Z(x). \tag{7}$$

This equation reveals a crucial insight: the log-probability gain of the optimal policy over the reference is determined by the reward $\mathcal{R}(x, y)$ offset by a per-prompt normalization term $\log Z(x)$. This term acts as an input-dependent baseline reward: only samples with rewards above this threshold are assigned higher probability by the optimal policy.

As proven in Appendix A, this policy ratio is strictly increasing with respect to the reward $\mathcal{R}(x, y)$. This naturally defines an ideal classification rule for whether a response $y$ should be preferred over the reference policy's distribution:

$$\pi^*(y|x) > \pi_{\text{ref}}(y|x) \quad \Longleftrightarrow \quad \mathcal{R}(x, y) > \beta \log Z(x), \tag{8}$$

and similarly, $\pi^*(y|x) < \pi_{\text{ref}}(y|x)$ when $\mathcal{R}(x, y) < \beta \log Z(x)$. This suggests that we could, in principle, train a policy by classifying samples as "preferred" or "dispreferred" relative to the reference policy. However, this is intractable because the ideal decision boundary, $\tau^*(x) = \beta \log Z(x)$, is instance-dependent and relies on the partition function $Z(x)$ we seek to avoid.

### 2.2.2 A Tractable Proxy: Thresholding as Pseudo-Preference Generation

We replace the intractable ideal rule with a computable proxy with two reasonable assumptions based on the available data $\mathcal{D} = \{(x_i, y_i, s_i)\}$. First, we treat the observed scalar score $s$ as a proxy for the true latent reward $\mathcal{R}(x, y)$. Second, we approximate the instance-specific ideal threshold $\tau^*(x)$ with a global threshold $\tau$, calculated as an empirical quantile (*e.g.*, the median) of scores. By substituting these components into the ideal inequality, we derive UPO's pseudo-preference decision rule:

$$(x, y, s) \mapsto \begin{cases} \pi^*(y|x) > \pi_{\text{ref}}(y|x) & \text{(Pseudo-Preferred)} & \text{if } s \geq \tau \\ \pi^*(y|x) < \pi_{\text{ref}}(y|x) & \text{(Pseudo-Rejected)} & \text{if } s < \tau \end{cases} \tag{9}$$

By generating a binary pseudo-label $l = \mathbb{F}[s \geq \tau]$ for each sample, UPO effectively classifies it as "preferred" or "dispreferred" within the context of its batch.

### 2.2.3 THE UPO OBJECTIVE AND LOSS FUNCTION

By combining the above tractable proxies, we formalize the UPO objective as a weighted binary cross-entropy loss. For a given sample $(x, y, s)$, we aim to align the sign of our implicit policy score $\hat{s}_{\theta,\mathrm{ref}}$ with a pseudo-label $l$ derived from the scalar score $s$.

$$\mathcal{L}_{\mathrm{UPO}} = -\mathbb{E}_{(x,y,s)\sim\mathcal{D}}\left[w(s, \tau)\left(l \log \sigma(\hat{s}_{\theta,\mathrm{ref}}) + (1 - l)\log(1 - \sigma(\hat{s}_{\theta,\mathrm{ref}}))\right)\right] \quad (10)$$

where $\sigma(\cdot)$ is the logistic function and the components are defined as follows:

**Implicit Policy Score** $\hat{s}_{\theta,\mathrm{ref}}$. This is the core quantity being optimized, representing the log-ratio of the learned policy's likelihood to the reference policy's, scaled by $\beta$.

$$\hat{s}_{\theta,\mathrm{ref}}(x, y) = \beta \log \frac{\pi_\theta(y|x)}{\pi_{\mathrm{ref}}(y|x)}.$$

The loss encourages $\hat{s}_\theta$ to be positive for preferred samples and negative for dispreferred ones.

**Pseudo-Label** $l$. The ground-truth label for the classification task is determined by comparing the sample's score $s$ against the global threshold $\tau$.

$$l = \mathbb{K}[s \geq \tau],$$

where $\mathbb{K}[\cdot]$ is the indicator function. This effectively converts the continuous score $s$ into a binary preference signal.

**Global Threshold** $\tau$. To create an adaptive decision boundary, we set $\tau$ as the $p$-th percentile of scores within each batch (or epoch), *e.g.*, $\tau = \mathrm{percentile}(\{s_j\}, p = 50)$.

**Confidence Weighting** $w(s, \tau)$. Intuitively, samples with scores far from the decision boundary provide a stronger, less ambiguous training signal. We incorporate this by weighting each sample's loss contribution based on its distance to the threshold:

$$w(s, \tau) = 1 + c \cdot |s - \tau|,$$

where $c$ is a hyperparameter scaling the weighting effect. This prioritizes high-confidence samples, stabilizing training and focusing the model on clear-cut cases.

By minimizing $\mathcal{L}_{\mathrm{UPO}}$, we guide the policy $\pi_\theta$ to assign higher relative log-probabilities to samples deemed desirable by the scalar feedback $s$, this provides a robust and effective method for policy alignment from unpaired data.

### 2.3 THEORETICAL ANALYSIS

Although UPO is motivated by practical considerations, it is crucial to ensure its statistical soundness. The KL-optimal decision rule states that the probability of a sample $y$ should be increased if and only if its reward exceeds the oracle baseline $\tau^*(x) = \beta \log Z(x)$, which is intractable to compute. UPO replaces this baseline with a tractable proxy $\tau$ derived from the empirical score distribution, together with a confidence-weighting scheme $w(s, \tau)$. Our theoretical analysis, detailed in Appendix B, provides strong guarantees for this approach.

**Theorem 2.1** (UPO Guarantees, Informal). *Let $\hat{\theta}_n$ be the minimizer of the empirical UPO loss under mild regularity conditions (see Appendix B). Then:*

1. ***Consistency.*** $\hat{\theta}_n \to \theta^*$ *as* $n \to \infty$, *i.e., UPO recovers the population optimum.*

2. ***Controlled Bias.*** *The estimator is asymptotically unbiased, with leading-order error* $\mathbb{E}[\hat{\theta}_n] - \theta^* = O(1/n)$ *determined by the curvature* ($H$), *variance* ($S$), *and asymmetry* ($J$) *of the UPO loss.*

3. ***Calibration.*** *The UPO surrogate classifier, based on the empirical threshold $\tau$, aligns with the KL-optimal rule $\mathbb{K}[R(x, y) > \tau^*(x)]$ up to quantifiable estimation error.*

These results provide a principled foundation for UPO. The estimator is statistically consistent, and any bias vanishes at the standard $O(1/n)$ rate, ensuring reliability on large datasets. Moreover, the calibration guarantee formally justifies our key design choice: empirical quantile thresholding yields pseudo-preference labels that faithfully approximate the intractable KL-optimal rule. Together, these properties establish UPO as a theoretically grounded framework rather than a heuristic.

## 2.4 IMPLEMENTATION AND ALGORITHMIC DETAILS

### 2.4.1 LOG-LIKELIHOOD COMPUTATION

The computation of the log-likelihood term $\log \pi_\theta(y|x)$ in our objective depends on the paradigm of the generative model. We detail the approaches for continuous-space diffusion models and discrete-space MaskGIT models.

**Diffusion Models (Continuous Outputs).** For diffusion-based generative models, the exact log-likelihood $\log \pi_\theta(y|x)$ is generally intractable. Following prior work on preference optimization for diffusion (Lee et al., 2023; Wallace et al., 2024), we approximate the log-likelihood under a Gaussian observation model. The diffusion training loss (Ho et al., 2020) minimizes the reconstruction error, which corresponds to the negative log-likelihood of a Gaussian with fixed variance:

$$p(y|x) = \mathcal{N}(\hat{y}_\theta(x), \sigma^2 I) \quad \Rightarrow \quad \log p(y|x) = -\frac{1}{2\sigma^2}\|y - \hat{y}_\theta(x)\|^2 + \text{const.} \tag{11}$$

Thus, we use the scaled negative mean squared error as a surrogate for the log-likelihood:

$$\log \pi_\theta(y|x) \approx -\frac{1}{T} \cdot \text{MSE}(y, \hat{y}_\theta(x)),$$

where $\hat{y}_\theta(x)$ is the one-step denoised prediction and $T$ is a temperature hyperparameter controlling the approximation scale.

**MaskGIT (Discrete Outputs).** MaskGIT (Chang et al., 2022) operates in a discrete token space, where an image $y$ is represented as $N$ tokens $(t_1, \ldots, t_N)$ from a VQ-GAN encoder (Esser et al., 2021). Its training objective is to predict masked tokens given the visible context and condition $x$. In this case, the log-likelihood is directly computable as the sum of log-probabilities at the masked positions:

$$\log \pi_\theta(y|x) = \frac{1}{|M|} \sum_{i \in M} \log p_\theta(t_i \mid y_{\backslash M}, x),$$

where $M$ is the set of masked indices and $y_{\backslash M}$ denotes the unmasked tokens. We normalize by $|M|$ to ensure comparability across samples with different mask sizes.

### 2.4.2 ALGORITHMIC DETAILS

We present two variants of UPO. The primary approach, Offline UPO, operates on a fixed dataset $\mathcal{D} = \{(x_i, y_i)\}$. Typically, this dataset is first created by generating outputs $y_i$ using the initial policy $\pi_\theta$ for a given set of prompts $\{x_i\}$. Reward scores are precomputed for this dataset, and a global threshold $\tau$ is estimated once per epoch. In contrast, Online UPO computes rewards on-the-fly with a memory bank, allowing adaptation to distributional shift. However, this requires repeated sampling and reward evaluation, which is computationally demanding. For this reason, offline UPO serves as the default choice in practice. The step-by-step procedure for offline UPO is given in Algorithm 1, while the online variant is detailed in Appendix C.

In production-scale training, reward scores for massive datasets are often available, but their distribution may not match that of the policy $\pi_\theta$ under training. Consequently, these scores cannot be directly used to define thresholds. To address this, we construct smaller proxy sets by sampling prompts, generating outputs from $\pi_{ref}$, and scoring them with the reward model. The thresholds $\tau$ estimated from these proxy sets can then be transferred to the much larger pre-scored dataset. As more proxy sets are accumulated, the estimate of $\tau$ becomes increasingly reliable. As established in our theoretical analysis (Theorem 2.1), the asymptotic bias vanishes at rate $O(1/n)$, ensuring that thresholds estimated from sufficiently large proxy sets generalize robustly to production-scale datasets.

---

**Algorithm 1** Unpaired Preference Optimization (UPO)

---

**Require:** Initial policy $\pi_\theta$, reward model $r(\cdot)$, dataset $\mathcal{D} = \{(x_i, y_i)\}$, batch size $B$, threshold percentile $p$, scaling coefficient $c$, temperature $\beta$

1: Initialize reference policy: $\pi_{\text{ref}} \leftarrow \pi_\theta$          ▷ Frozen copy at initialization
2: Compute all reward scores $s_i = r(x_i, y_i)$ for $(x_i, y_i) \in \mathcal{D}$
3: Compute global threshold: $\tau = \text{percentile}(\{s_i\}, p)$
4: **for** each training epoch **do**
5:      **for** each batch $\{(x_j, y_j, s_j)\}_{j=1}^B \sim \mathcal{D}$ **do**
6:          **for** each sample $(x_j, y_j, s_j)$ in the batch **do**
7:              Compute pseudo-label $l = \mathbb{1}[s_j \geq \tau]$
8:              Compute weight $w = 1 + c \cdot |s_j - \tau|$
9:              Compute implicit score $\hat{r}_j = \beta \cdot \left[ \log \pi_\theta(y_j|x_j) - \log \pi_{\text{ref}}(y_j|x_j) \right]$
10:            Compute per-sample loss $\ell_j = -w \cdot [l \log \sigma(\hat{r}_j) + (1 - l) \log(1 - \sigma(\hat{r}_j))]$
11:          **end for**
12:          $\mathcal{L} = \frac{1}{B} \sum_{j=1}^B \ell_j$
13:          Update $\pi_\theta \leftarrow \text{GradientStep}(\pi_\theta, \nabla_\theta \mathcal{L})$
14:      **end for**
15:      Update reference policy: $\pi_{\text{ref}} \leftarrow \pi_\theta$
16: **end for**

---

## 3 EXPERIMENTS

### 3.1 TEXT-TO-IMAGE SYNTHESIS WITH UPO

We collect 10,000 high-quality prompts, termed *MeiPrompts*, for both supervised fine-tuning (SFT) and unpaired performance optimization (UPO). Figure 2 presents an analysis of the prompt set, including (a) the prompt length distribution, (b) the most frequent keywords, and (c) a CLIP-based t-SNE visualization comparing the semantic space of MeiPrompts (train set) and HPS Prompts (test set). The clear distributional divergence indicates no data leakage between the training and test sets.

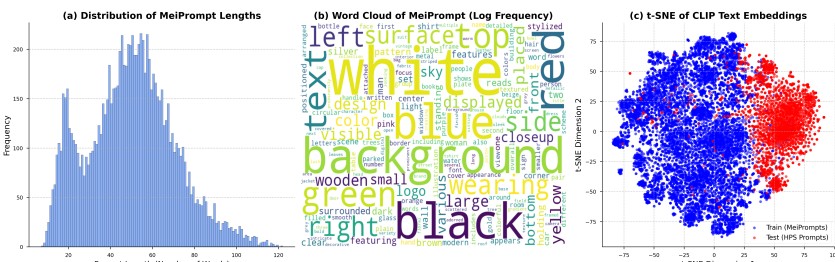

Figure 2: (a) Prompt length distribution, (b) word cloud, and (c) CLIP+tSNE semantic visualization of MeiPrompts and HPS Prompts.

We conduct text-to-image experiments using two foundation models: Stable Diffusion v1.4 (Rombach et al., 2022) and Meissonic (Bai et al., 2024). First, each model is used to generate images conditioned on MeiPrompts. Second, we score these generated image-text pairs using four reward models: HPSv2.1 (Wu et al., 2023), PickScore (Kirstain et al., 2023), ImageReward (Xu et al., 2023) and Laion Aesthetic Score (Schuhmann et al., 2022), and obtain their median values as threshold $\tau$. Third, for supervised finetuning, we finetune the original model with generated image-text pairs whose reward scores above the threshold $\tau$, and for unpaired preference optimization, we apply the unpaired optimization method introduced in the previous section. To ensure fairness, both SFT and UPO are trained with identical hyperparameters: batch size (128), training steps (78), and learning rate (1e-5). We set $\beta = 1$, $T = 0.001$ and $c = 5$ in UPO loss function (Eq. 10) and present ablations in the subsequent section.

We report both the mean and median scores on HPS Prompts (Wu et al., 2023) across the four reward models in Table 1, we also visualize the full score distributions for Stable Diffusion v1.4 in Figure 5. Besides, we present qualitative comparisons between SFT and UPO on HPSv2.1 rewarding for stable diffusion v1.4 in Figure 3. For a more comprehensive comparison, we adopt GPT-5 as an automated judge for the full 3,200 image pairs in Figure 4.

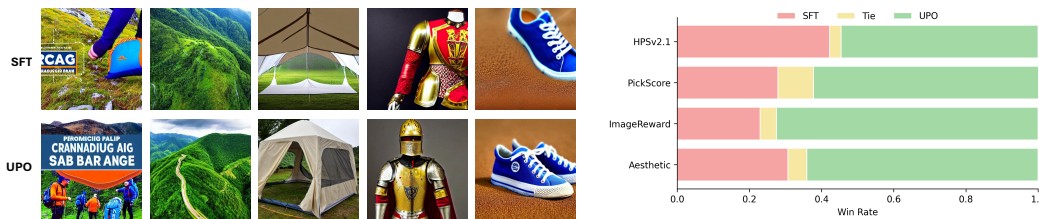

Figure 3: Qualitative comparisons between SFT and UPO for SD v1.4.

Figure 4: Win rate between SFT and UPO for SD v1.4.

Table 1: Quantitative comparison of text-to-image generation across original, supervised fine-tuned (SFT), and unpaired performance optimized (UPO) methods. Higher is better. SD v1.4 denotes Stable Diffusion v1.4; Evaluation is conducted on HPS Prompts using four different reward models.

| Model | Version | HPSv2.1 ($\uparrow$) | | PickScore ($\uparrow$) | | ImageReward ($\uparrow$) | | Aesthetic ($\uparrow$) | |
|---|---|---|---|---|---|---|---|---|---|
| | | Mean | Median | Mean | Median | Mean | Median | Mean | Median |
| SD v1.4 | Original | 0.2454 | 0.2462 | 20.8040 | 20.7784 | 0.1406 | 0.1773 | 5.4277 | 5.4293 |
| | +SFT | 0.2506 | 0.2520 | 20.7217 | 20.7006 | 0.2348 | 0.2870 | 5.4927 | 5.4948 |
| | +UPO | **0.2618** | **0.2631** | **20.9001** | **20.8907** | **0.3523** | **0.4246** | **5.6036** | **5.6122** |
| Meissonic | Original | 0.2810 | 0.2837 | 21.8315 | 21.7686 | 0.8230 | 0.9674 | 5.7692 | 5.7578 |
| | +SFT | 0.2912 | 0.2928 | 21.9105 | 21.8419 | 0.9215 | 1.0985 | 5.8013 | 5.7999 |
| | +UPO | **0.2915** | **0.2934** | **21.9421** | **21.8946** | **0.9369** | **1.1233** | **5.8270** | **5.8234** |

From both the quantitative and qualitative results, we observe that UPO consistently surpasses SFT for four reward models in most cases. This demonstrates the effectiveness of our unpaired preference optimization method in improving text-to-image alignment without requiring supervision from paired preference data.

## 3.2 TEXT-TO-VIDEO SYNTHESIS WITH UPO

We present text-to-video experiments on Wan 1.3B (Wan et al., 2025) in Appendix E.

## 3.3 ONLINE UPO

We present UPO and Online UPO comparisons in Appendix D.

## 3.4 ABLATION STUDY

We ablate the key hyperparameters of UPO, including the temperature $T$ used to approximate log-probability, the difference scaling factor $c$, and the preference strength coefficient $\beta$. All experiments are conducted by fine-tuning Stable Diffusion v1.4 with MeiPrompt and evaluating with MeiPrompt on the median HPSv2.1 score.

### 3.4.1 EFFECT OF TEMPERATURE $T$

The temperature $T$ scales the negative MSE used to approximate $\log \pi_\theta(y|x)$. As shown in Table 2a, a moderate temperature ($T = 0.001$) achieves the highest score. When $T$ is too large (*e.g.*, 0.1 or 1.0), the distribution over preferences becomes nearly uniform, causing instability and sharp performance degradation.

### 3.4.2 EFFECT OF DIFFERENCE SCALING FACTOR $c$

The factor $c$ amplifies the absolute difference between reward and threshold $\tau$, thereby increasing the weight of confident preferences. Table 2b shows that performance improves as $c$ increases from 2 to 5, but further growth (*e.g.*, $c = 20$) provides no consistent gain. Setting $c = 5$ offers a good balance.

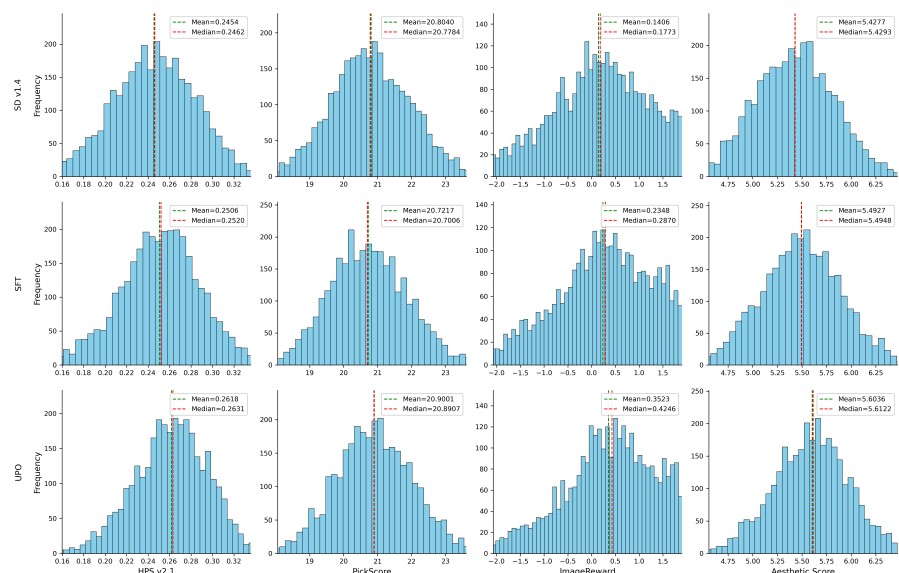

Figure 5: Score distributions by model and reward metric for SD v1.4.

| (a) Temperature $T$ | | | (b) Scaling Ratio $c$ | | | (c) Preference Strength $\beta$ | |
|---|---|---|---|---|---|---|---|
| **T** | **Median** | | **c** | **Median** | | **$\beta$** | **Median** |
| 0.00001 | 0.2378 | | 1 | 0.2478 | | 0.5 | 0.2469 |
| 0.0001 | 0.2438 | | 2 | 0.2480 | | 1.0 | **0.2526** |
| 0.001 | **0.2526** | | 5 | **0.2526** | | 2.0 | 0.2481 |
| 0.01 | 0.2319 | | 10 | 0.2481 | | | |
| 0.1 | 0.1411 | | 20 | 0.2482 | | | |

Table 2: Ablation on UPO hyperparameters using SD v1.4 fine-tuned with MeiPrompt. Reported values are median HPSv2.1 scores.

### 3.4.3  EFFECT OF PREFERENCE STRENGTH $\beta$

The coefficient $\beta$ controls the sharpness of the log-probability ratio in the UPO loss. As Table 2c illustrates, $\beta = 1.0$ works well across experiments, while larger or smaller values do not confer additional benefits.

Overall, the best UPO configuration for SD v1.4 is $\beta = 1$, $c = 5$, and $T = 0.001$. While other foundation models or reward models may work better with different values, this setting provides a default starting point for practice.

## 4  CONCLUSION

In this work, we introduced Unpaired Preference Optimization (UPO), a framework that extends direct preference optimization to settings without paired comparisons, enabling alignment directly from unpaired scalar scores. Our derivation revisits the KL-regularized objective and reveals an ideal but intractable decision rule governed by an instance-dependent oracle baseline. UPO provides a principled and tractable proxy to this rule via global thresholding and confidence-weighting mechanisms. Extensive experiments demonstrate that UPO consistently improves alignment of generative models, outperforming supervised fine-tuning across diverse settings. Our analysis further shows that the proxy becomes increasingly reliable as more scored samples are used to estimate the reward distribution, approaching the oracle decision rule in the large-data limit. Taken together, UPO establishes a flexible, efficient, and theoretically grounded approach to aligning generative models with unpaired real world human values.

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

# APPENDIX

## A PROOF OF MONOTONICITY FOR THE POLICY RATIO

**Theorem A.1** (Monotonicity of the Policy Ratio). *Let the optimal policy $\pi^*(y|x)$ be defined by the KL-regularized objective:*

$$\pi^*(y|x) = \frac{1}{Z(x)} \pi_{\text{ref}}(y|x) \exp\left(\frac{1}{\beta} r(x,y)\right), \tag{12}$$

*where the partition function $Z(x) = \sum_{y' \in \mathcal{Y}} \pi_{\text{ref}}(y'|x) \exp\left(\frac{1}{\beta} r(x,y')\right)$. Then, for any $y_k \in \mathcal{Y}$ such that $\pi_{\text{ref}}(y_k|x) > 0$, the ratio $\frac{\pi^*(y_k|x)}{\pi_{\text{ref}}(y_k|x)}$ is a strictly increasing function of its reward $r(x, y_k)$, provided there exists at least one alternative response $y' \neq y_k$ with $\pi_{\text{ref}}(y'|x) > 0$.*

*Proof.* Fix $y_k \in \mathcal{Y}$ and define $r := r(x, y_k)$ to simplify notation. We analyze the function:

$$f(r) := \frac{\pi^*(y_k|x)}{\pi_{\text{ref}}(y_k|x)} = \frac{\exp(r/\beta)}{Z(x)}, \tag{13}$$

where the normalization constant $Z(x)$ depends on $r$, since $r(x, y_k)$ is one of the terms in its summation.

To determine if $f(r)$ is strictly increasing, we compute its derivative with respect to $r$ using the quotient rule. Let $u(r) := \exp(r/\beta)$ and $v(r) := Z(x)$. Then $\frac{df}{dr} = \frac{u'v - uv'}{v^2}$.

The derivatives of $u(r)$ and $v(r)$ are:

$$u'(r) = \frac{1}{\beta} \exp(r/\beta), \tag{14}$$

$$v'(r) = \frac{d}{dr}\left[\sum_{y' \in \mathcal{Y}} \pi_{\text{ref}}(y'|x) \exp\left(\frac{r(x,y')}{\beta}\right)\right] = \pi_{\text{ref}}(y_k|x) \cdot \frac{1}{\beta} \exp(r/\beta). \tag{15}$$

The derivative $v'(r)$ only contains the term corresponding to $y_k$ because all other rewards $r(x, y')$ for $y' \neq y_k$ are treated as constants with respect to $r$.

Substituting these into the quotient rule expression:

$$\frac{df}{dr} = \frac{\left(\frac{1}{\beta} \exp(r/\beta)\right) Z(x) - \exp(r/\beta) \left(\pi_{\text{ref}}(y_k|x) \cdot \frac{1}{\beta} \exp(r/\beta)\right)}{Z(x)^2} \tag{16}$$

$$= \frac{\exp(r/\beta)}{\beta Z(x)^2} \cdot \left[Z(x) - \pi_{\text{ref}}(y_k|x) \exp(r/\beta)\right]. \tag{17}$$

The term in the brackets simplifies to:

$$Z(x) - \pi_{\text{ref}}(y_k|x)\exp(r/\beta) = \left(\sum_{y' \in \mathcal{Y}} \pi_{\text{ref}}(y'|x) \exp\left(\frac{r(x,y')}{\beta}\right)\right) - \pi_{\text{ref}}(y_k|x) \exp\left(\frac{r}{\beta}\right) \tag{18}$$

$$= \sum_{y' \neq y_k} \pi_{\text{ref}}(y'|x) \exp\left(\frac{r(x,y')}{\beta}\right). \tag{19}$$

Since $\pi_{\text{ref}}(y'|x) \geq 0$ and $\exp(\cdot) > 0$, each term in this sum is non-negative. By the theorem's condition, there is at least one $y' \neq y_k$ with $\pi_{\text{ref}}(y'|x) > 0$, so this sum is strictly positive.

Therefore, the derivative in Eq. 17 is a product of strictly positive terms:

$$\frac{df}{dr} = \underbrace{\frac{\exp(r/\beta)}{\beta Z(x)^2}}_{>0} \cdot \underbrace{\left(\sum_{y' \neq y_k} \pi_{\text{ref}}(y'|x) \exp\left(\frac{r(x,y')}{\beta}\right)\right)}_{>0} > 0. \tag{20}$$

Since the derivative is strictly positive, the function $f(r)$ is strictly increasing in $r$. $\qquad\square$

# B PROOFS OF UPO GUARANTEES

This appendix provides the formal assumptions, theorems, and proofs for the guarantees of UPO summarized in Theorem 2.1 of the main text.

## B.1 ASSUMPTIONS

**Assumption B.1** (Regularity Conditions for UPO). *Let $\ell(\theta; z)$ denote the per-sample UPO loss. Assume:*

1. *(**Identifiability**) The population loss $L(\theta) = \mathbb{E}_z[\ell(\theta; z)]$ has a unique minimizer $\theta^*$ in an open neighborhood $\mathcal{N}$.*

2. *(**Smoothness**) $\ell(\theta; z)$ is three-times continuously differentiable in $\mathcal{N}$ almost surely. The population derivatives $\nabla^k L(\theta)$ for $k = 1, 2, 3$ exist and are continuous at $\theta^*$.*

3. *(**Regularity**) The Hessian $H = \nabla^2 L(\theta^*)$ is positive definite. The score $\nabla\ell(\theta^*; z)$ has finite second moments with covariance $S = \mathrm{Cov}(\nabla\ell(\theta^*; z))$. A Central Limit Theorem holds for $\sqrt{n}\,\nabla L_n(\theta^*)$.*

## B.2 CONSISTENCY OF THE UPO ESTIMATOR

**Corollary B.2** (Consistency of UPO). *Under Assumption B.1, the UPO estimator*

$$\hat{\theta}_n = \arg\min_\theta L_n(\theta), \quad L_n(\theta) = \tfrac{1}{n}\sum_{i=1}^n \ell(\theta; z_i)$$

*is consistent: $\hat{\theta}_n \xrightarrow{p} \theta^*$ as $n \to \infty$.*

*Proof.* This follows directly from standard M-estimation theory: $L_n(\theta)$ converges uniformly to $L(\theta)$, which has a unique minimizer $\theta^*$ under Assumption B.1. The argmin consistency theorem then yields $\hat{\theta}_n \xrightarrow{p} \theta^*$. $\square$

## B.3 ASYMPTOTIC BIAS OF THE UPO ESTIMATOR

**Theorem B.3** (Asymptotic Bias of UPO). *Under Assumption B.1, the expectation of $\hat{\theta}_n$ satisfies*

$$\mathbb{E}[\hat{\theta}_n] - \theta^* = \tfrac{1}{n}B_1(\theta^*) + o(1/n),$$

*where*

$$(B_1(\theta^*))_a = -\tfrac{1}{2}H_{ab}^{-1}\,J_{bcd}\,(H^{-1}SH^{-1})_{cd}, \tag{21}$$

*with $H = \nabla^2 L(\theta^*)$, $S = \mathrm{Cov}(\nabla\ell(\theta^*; z))$, and $J_{bcd} = \mathbb{E}[\partial^3\ell(\theta^*; z)/\partial\theta_b\partial\theta_c\partial\theta_d]$.*

*Proof.* **Step 1: First-order condition and Taylor expansion.** By optimality, $0 = \nabla L_n(\hat{\theta}_n)$. Expanding around $\theta^*$ gives

$$0 = \nabla L_n(\theta^*) + \nabla^2 L_n(\theta^*)(\hat{\theta}_n - \theta^*) + \tfrac{1}{2}\nabla^3 L_n(\bar{\theta})[\hat{\theta}_n - \theta^*, \hat{\theta}_n - \theta^*] + r_n, \tag{22}$$

where $\bar{\theta}$ lies between $\hat{\theta}_n$ and $\theta^*$, and $r_n = o_p(\|\hat{\theta}_n - \theta^*\|^2) = o_p(n^{-1})$.

**Step 2: Isolate $\Delta = \hat{\theta}_n - \theta^*$.** Rearranging Eq. 22:

$$\Delta = -\left[\nabla^2 L_n(\theta^*)\right]^{-1}\nabla L_n(\theta^*) - \tfrac{1}{2}\left[\nabla^2 L_n(\theta^*)\right]^{-1}\nabla^3 L_n(\bar{\theta})[\Delta, \Delta] + o_p(n^{-1}). \tag{23}$$

**Step 3: Take expectations.** Since $\mathbb{E}[\nabla L_n(\theta^*)] = 0$ and $\nabla^2 L_n(\theta^*) \xrightarrow{p} H$, we replace random Hessian and third derivatives by $H$ and $J$ up to $o(n^{-1})$ terms:

$$\mathbb{E}[\Delta] = -\tfrac{1}{2}H^{-1}J\,\mathbb{E}[\Delta \otimes \Delta] + o(n^{-1}). \tag{24}$$

**Step 4: Insert asymptotic covariance.** From standard M-estimator theory,

$$\mathbb{E}[\Delta \otimes \Delta] = \frac{1}{n} H^{-1} S H^{-1} + o(n^{-1}). \tag{25}$$

Substituting Eq. 25 into Eq. 24 gives

$$\mathbb{E}[\Delta] = -\frac{1}{2n} H^{-1} J \big( H^{-1} S H^{-1} \big) + o(n^{-1}),$$

which matches Eq. 21. $\qquad\square$

**Remark.** If $\ell(\theta; z)$ is the negative log-likelihood of a correctly specified model, then $S = H = I(\theta^*)$ (Fisher information), further simplifying the bias term.

### B.4 CALIBRATION OF UPO PSEUDO-LABELS

**Proposition B.4** (Calibration of UPO). *Let $\tau^*(x) = \beta \log Z(x)$ be the KL-optimal baseline. Suppose scores satisfy $s = g(R(x, y)) + \xi$, where $g$ is strictly increasing and $\xi$ is sub-Gaussian. If $\tau$ is estimated as the empirical $p$-quantile with error $\varepsilon_\tau = O(1/\sqrt{n})$, then*

$$\Pr\big[l \neq l^*\big] = O(\varepsilon_\tau + \|\xi\|_{\psi_2}),$$

*where $l = \mathbb{K}[s \geq \tau]$ and $l^* = \mathbb{K}[R(x, y) \geq \tau^*(x)]$.*

*Proof.* By Theorem A.1 in Appendix A, the KL-optimal rule reduces to thresholding $R(x, y)$ against $\tau^*(x)$. Since $g$ preserves order, classification by $s$ matches that by $R$ up to noise $\xi$. The empirical quantile $\tau$ concentrates around the true quantile, so label flips occur only if $\xi$ or $\varepsilon_\tau$ is large enough to cross the boundary, yielding the stated bound. $\qquad\square$

## C THE ALGORITHM PROCEDURE OF ONLINE UPO

---

**Algorithm 2** Online Unpaired Preference Optimization (Online-UPO)

---

**Require:** Initial policy $\pi_\theta$, Prompt-only dataset $\mathcal{X} = \{x_i\}$, reward model $r(\cdot)$, memory bank $\mathcal{M}$, memory size $M$, threshold percentile $p$, scaling coefficient $c$, temperature $\beta$, batch size $B$
1:  Initialize reference policy $\pi_{\text{ref}} \leftarrow \pi_\theta$
2:  Initialize memory bank $\mathcal{M} \leftarrow \emptyset$
3:  **for** each training epoch **do**
4:      **for** each batch of prompts $\{x_j\}_{j=1}^B \sim \mathcal{X}$ **do**
5:          Sample outputs $y_j \sim \pi_{\text{ref}}(\cdot|x_j)$
6:          Compute rewards $s_j = r(x_j, y_j)$
7:          Update memory bank $\mathcal{M} \leftarrow \text{FIFO\_Update}(\mathcal{M}, \{s_j\})$
8:          **if** $|\mathcal{M}| < M/2$ **then**
9:              **continue** $\qquad\qquad\qquad\qquad\qquad$ ▷ Delay updates until memory bank is warm
10:         **end if**
11:         Compute threshold $\tau = \text{percentile}(\mathcal{M}, p)$
12:         **for** each sample $(x_j, y_j, s_j)$ in the batch **do**
13:             Compute pseudo-label $l = \mathbb{K}[s_j \geq \tau]$
14:             Compute weight $w = 1 + c \cdot |s_j - \tau|$
15:             Compute implicit score $\hat{r}_j = \beta \cdot [\log \pi_\theta(y_j|x_j) - \log \pi_{\text{ref}}(y_j|x_j)]$
16:             Compute per-sample loss $\ell_j = -w \cdot [l \log \sigma(\hat{r}_j) + (1 - l) \log(1 - \sigma(\hat{r}_j))]$
17:         **end for**
18:         Compute batch loss: $\mathcal{L} = \frac{1}{B} \sum_{j=1}^B \ell_j$
19:         Update policy: $\pi_\theta \leftarrow \text{GradientStep}(\pi_\theta, \nabla_\theta \mathcal{L})$ $\quad$ ▷ Train current policy on data from ref
20:     **end for**
21:     Update reference policy: $\pi_{\text{ref}} \leftarrow \pi_\theta$
22: **end for**

---

Online UPO is an adaptive variant of UPO, designed for scenarios where the data distribution may significantly shift during training. Instead of using a pre-computed dataset, it generates samples and

Table 3: Comparison between standard UPO and Online UPO under the HPSv2.1 metric. All models are trained and evaluated with MeiPrompt.

| Method | $c$ | $T$ | $\beta$ | Median HPSv2.1 |
|--------|-----|-----|---------|----------------|
| Original SD1.4 | - | - | - | 0.2364 |
| SFT | - | - | - | 0.2426 |
| UPO (v1) | 10 | 0.001 | 1 | 0.2443 |
| Online UPO (v1) | 10 | 0.001 | 1 | 0.2355 |
| UPO (v2) | 20 | 0.001 | 1 | 0.2482 |
| Online UPO (v2) | 20 | 0.001 | 1 | 0.2449 |
| UPO (v3, best) | 5 | 0.001 | 1 | **0.2526** |
| Online UPO (v3) | 5 | 0.001 | 1 | 0.2381 |

computes rewards on-the-fly at each step. A memory bank $\mathcal{M}$ stores recent reward scores, allowing the decision threshold $\tau$ to be dynamically re-estimated. This adaptability comes at a significant computational cost, making the offline version (Algorithm 1) the more practical choice for most large-scale applications. The detailed procedure is provided in Algorithm 2.

# D  ONLINE UPO

We compare UPO and Online UPO on the HPSv2.1 metric in Table 3. All models are fine-tuned on Stable Diffusion v1.4 using MeiPrompt and evaluated on the same prompt set. We report the median score to assess performance.

Across all configurations, Online UPO performs slightly worse than the offline variant. Two factors appear central. First, Online UPO estimates the reward threshold $\tau$ on the fly from a memory bank of 1024 samples. This introduces higher variance and occasional inaccuracies compared to the offline method, which computes $\tau$ once from the full reward distribution (10k samples). Second, Online UPO requires image generation and reward evaluation during training, leading to roughly $10\times$ higher wall-clock cost.

Overall, under our setup the offline variant is both more stable and more efficient.

# E  TEXT-TO-VIDEO SYNTHESIS WITH UPO

We extend our study to text-to-video generation. To this end, we collect 15,218 high-quality prompts, denoted as *MeiPrompts-V*, and split them into training and test sets with an 8:2 ratio. We randomly subsample a portion of the dataset for our experiments. We adopt Wan 1.3B (Wan et al., 2025) as the foundation model and VideoReward (Liu et al., 2025b) as the reward model.

Table 4 reports results on the VideoAlign benchmark, including VQ (visual quality), MQ (motion quality), TA (temporal alignment), and the overall score. UPO improves over supervised fine-tuning (SFT-LoRA) in most cases.

Table 4: Text-to-video results on the VideoAlign benchmark. UPO-LoRA improves over SFT-LoRA in most cases.

| Method | VQ Score | MQ Score | TA Score | Overall Score |
|--------|----------|----------|----------|---------------|
| Original | -0.7963 | -0.4316 | -0.8639 | -2.0918 |
| SFT-LoRA | -0.6054 | -0.4159 | **-0.6705** | -1.6918 |
| UPO-LoRA | **-0.5631** | **0.0627** | -1.0753 | **-1.5757** |

# F  RELATED WORK

## F.1  GENERATIVE MODELS.

Generative modeling has witnessed a rapid evolution, from Generative Adversarial Networks (GANs) (Goodfellow et al., 2020) and Variational Autoencoders (VAEs) (Kingma et al., 2013), to the current dominance of diffusion models (Sohl-Dickstein et al., 2015; Ho et al., 2020; Podell et al., 2023; Betker et al., 2023; Black-Forest-Labs, 2024). Denoising diffusion probabilistic models (DDPMs) have established a new state-of-the-art in high-fidelity image synthesis by iteratively reversing a noise-injection process. Their remarkable generative quality and training stability have made them the *de-facto* architecture for large-scale text-to-image systems.

A key breakthrough enabling granular control over the generation process was classifier-free guidance (Ho & Salimans, 2022), which allows for a trade-off between sample fidelity and diversity without needing an external classifier model. While classifier-free guidance provides a powerful mechanism for conditioning on explicit text prompts, it does not inherently address alignment with more abstract or ineffable human preferences, such as aesthetic appeal, compositional coherence, or stylistic nuance. This limitation necessitates a more direct approach to learn from human feedback.

## F.2  IMPROVING LANGUAGE MODELS USING PREFERENCE OPTIMIZATION.

The challenge of aligning powerful base models with human intent was first tackled systematically in the domain of large language models (LLMs). The seminal paradigm of Reinforcement Learning from Human Feedback (RLHF) (Christiano et al., 2017) demonstrated that models could be fine-tuned to align with complex human values. The canonical RLHF pipeline, famously used to train InstructGPT and ChatGPT (Ouyang et al., 2022), involves three stages: supervised fine-tuning (SFT), training a reward model (RM) on human preference labels, and then fine-tuning the SFT policy using reinforcement learning (*e.g.*, PPO (Schulman et al., 2017)) to maximize the learned reward.

Despite its success, RLHF is notoriously complex and unstable, requiring the training and orchestration of multiple models and inheriting the hyperparameter sensitivity of deep RL algorithms. This motivated a search for simpler, more direct alignment methods. Direct Preference Optimization (DPO) (Rafailov et al., 2023) emerged as an elegant and effective alternative. DPO reframes the preference learning problem as a simple binary classification task on pairs of preferred and rejected responses. By deriving a direct mapping from this loss to the optimal policy under a KL-divergence constraint, DPO bypasses the need for explicit reward modeling and unstable RL training, offering a more stable and efficient alignment procedure. This shift from explicit reward modeling to direct preference optimization represents a significant advance in the field.

Recent work has sought to generalize direct preference optimization beyond the constraint of paired data, enabling learning from more flexible feedback structures. Viewing the problem from a probabilistic inference perspective, Abdolmaleki et al. (2024) propose a framework that handles unpaired positive (accepted) and negative (rejected) examples, even when only one feedback type is available. Derived via an Expectation-Maximization (EM) approach, their method extends prior work by explicitly incorporating dis-preferred samples, resulting in an intuitive objective: maximizing the likelihood of preferred outcomes while minimizing that of dis-preferred ones, all regularized by a KL divergence to a reference policy. Addressing the same limitation from a different angle, Matrenok et al. (2025) introduce Quantile Reward Policy Optimization (QRPO) to enable offline policy fitting directly on absolute scalar rewards. Their key insight is that by transforming rewards into their quantiles, the resulting reward distribution becomes uniform. This masterstroke makes the otherwise intractable partition function $Z(x)$ analytically solvable, eliminating the need for preference pairs to cancel it out. The final algorithm learns via a simple regression objective, retaining the simplicity of policy fitting while leveraging the full information of absolute scores.

## F.3  IMPROVING VISION MODELS USING PREFERENCE OPTIMIZATION.

Inspired by successes in LLMs, preference-based alignment has been increasingly adapted for diffusion models (Lee et al., 2023; Yang et al., 2024b; Deng et al., 2024; Yang et al., 2024a; Li et al., 2024; Ren et al., 2025; Yang et al., 2025; Zhang et al., 2024). Early approaches mirrored

the RLHF paradigm, first training an explicit reward or aesthetic-scoring model from human judgments (Schuhmann et al., 2022) and then using reinforcement learning to fine-tune the diffusion process, as exemplified by DDPO (Black et al., 2023). While effective, these methods reintroduced the complexities and training instabilities inherent to RL. More recently, the conceptual elegance of DPO (Rafailov et al., 2023), optimizing a KL-regularized objective under the Bradley–Terry preference model, spurred the development of direct preference optimization for diffusion (Wallace et al., 2024). These methods adapt the DPO loss to the diffusion framework, directly fine-tuning text-to-image models on preference pairs to enhance qualities like aesthetics and prompt faithfulness without out the overhead of RL. Besides, Generalized Reinforcement Policy Optimization (GRPO) (Xue et al., 2025; Liu et al., 2025a) emerges as a variant of proximal policy optimization recently.

However, a common thread unites this entire line of work: a fundamental reliance on paired preference data. This constraint limits their applicability in scenarios where feedback is only available as unpaired, absolute scores, highlighting a critical gap that our work, unpaired preference optimization (UPO), is designed to address.

## G  PSEUDOCODE OF UPO

We provide PyTorch-style pseudocode for the UPO loss function. The implementation closely follows the formulation in Section 2.4, using log-likelihood ratios between the current policy and the reference model, reweighted by relative scores:

```python
import torch
import torch.nn.functional as F

def compute_upo_loss(log_probs, ref_log_probs, relative_scores,
                     beta=1.0, c=1.0):
    """Args:
        log_probs: log pi_theta(y|x), shape (B,)
        ref_log_probs: log pi_ref(y|x), shape (B,)
        relative_scores: r(y) - tau, shape (B,)
        beta: temperature for reward difference
        c: weight scaling for RM difference
    """
    # Implicit reward difference
    reward_diff = beta * (log_probs - ref_log_probs) # (B,)

    # Preference direction and confidence weighting
    signs = torch.sign(relative_scores) # (B,)
    weights = 1 + c * relative_scores.abs() # (B,)

    # Binary logistic loss (use log1p for stability)
    sig = torch.sigmoid(reward_diff)
    pos_loss = -weights * torch.log(sig + 1e-12)
    neg_loss = -weights * torch.log1p(-sig + 1e-12)

    loss = torch.where(signs >= 0, pos_loss, neg_loss)
    return loss.mean()
```

## H  THE USE OF LARGE LANGUAGE MODELS

During the preparation of this paper, large language models were used only for language polishing and minor editing. All research ideas, methods, and experimental results were carried out entirely by the human authors.

