# OpenReview forum: "Unpaired Preference Optimization: Aligning Visual Generative Models with Scalar Feedback"
_ICLR.cc/2026/Conference — ICLR 2026 Conference Withdrawn Submission_

### Official Review · Reviewer_EnpB · 2025-10-29

**Soundness:** 2
**Presentation:** 3
**Contribution:** 2
**Rating:** 6
**Confidence:** 4

**Summary:**

This paper introduce UPO for aligning visual generative models like diffusion and MaskGIT using unpaired scalar feedback, instead of paired preferences required by DPO. They derive it from KL-regularized objective, approximating the intractable baseline with global threshold from score distribution, and add confidence weighting. Expriments on stable diffusion and meissonic with various reward models show it outperform SFT baselines. Theoretical analysis provide guarantees on consistency and bias.

**Strengths:**

- The idea to extend DPO to unpaired data is timely and practical, since real-world feedback often come as scores not pairs. This could broaden application in image/videos generation.

- the derivation from the optimal KL rule is nice, this makes UPO feel less heuristic than some alternatives. The thresholding and weighting mechanisms seem intuitive and effective base on the ablations.

- empirically, the experiments are comprehensive, it covers different models (diffusion and MaskGIT) and reward functions (HPS, PickScore, etc.). Using GPT-4 as judge for qualitative comparison is ok.

- theoretical section is strong, with informal theorem on consistency and calibration.

**Weaknesses:**

The approximation of instance-dependent $tau^*(x)$ with global tau might loose some nuance, especially if score distributions vary a lot across prompts. Authors acknowledge this but no deep analysis on when it fails.

Offline UPO rely on proxy datasets for threshold estimation, which could introduce bias if proxy not representative. Online variant is mentioned but dismissed as too expensive without much empirical comparison.

Vision focus is good, but no comparison to language model alignment methods that handle unpaired data, like in recent works on reward modeling. Would be nice to see if UPO generalize there.

Some implementation details vague, e.g., how temperature T chosen for diffusion log-likelihood approximation? Ablations help, but more guidance needed for reproduction.

**Questions:**

see above

---

### Official Review · Reviewer_3QQ6 · 2025-10-29

**Soundness:** 2
**Presentation:** 2
**Contribution:** 2
**Rating:** 2
**Confidence:** 5

**Summary:**

RLHF and other preference optimization frameworks like DPO, IPO relies on the availability of preference datasets to align large language model. A lot of datasets have absolute scores for each sample instead of pairwise scores. The authors attempt propose an alignment methodology suited for aligning LLMs using absolute scores in such cases. By re-arranging the RLHF mathematical formulation, the authors show that the preference alignment frameworks attempts to increase the probability of preferred response if the reward of the response is above an intractable, per-sample threshold. They use the median of the score distribution as a proxy for the threshold to generate preferred and dis-preferred samples. Furthermore, the difference in the rewards serves as a sample weight for the mini-batch loss calculation.

Through empirical experiments, the authors show that this method works on Diffusion Model and MaskGT. The aligned models shows higher estimated rewards compared to SFT based models.

**Strengths:**

1. The problem that this work attempts to tackle is important. It is true that a lot of datasets have absolute scores for samples and generating pairwise scores is not straightforward.
2. The proposed methodology and the experiments are easy to understand.

**Weaknesses:**

1. The claim that the median of the score distribution serves as a principled proxy for the intractable partition function seems overstated.
2. Furthermore, it is not clear how using a threshold computed globally across all the prompts could be used for LLM alignment. For LLM alignment, the preferred and dispreferred samples come from a single prompt. The alignment methodologies attempt to increase the probability of the preferred response while decreasing the probability of dispreferred responses. It is not clear how this methodology fits in with the LLM alignment paradigm.
3. The empirical results does not have any baselines where other preference optimization frameworks are used. It is not clear how this work compares to the existing preference optimization methodologies. I would encourage the authors to run this experiment using standard datasets like Alpaca, MTBench etc. and compare the results to the existing methods.

**Questions:**

Please address the weaknesses.

---

### Official Review · Reviewer_Sn97 · 2025-10-30

**Soundness:** 2
**Presentation:** 2
**Contribution:** 1
**Rating:** 2
**Confidence:** 3

**Summary:**

This paper introduces Unpaired Preference Optimization (UPO), a method to align generative models using unpaired scalar feedback (e.g., 1-5 star ratings) rather than the paired comparisons required by methods like DPO. The core idea is to approximate the intractable, instance-dependent KL-optimal decision rule with a simple, global threshold (e.g., the median score of a batch). Samples above this threshold are labeled "pseudo-preferred" and those below "pseudo-rejected," transforming the problem into a binary classification task. The loss is also weighted by the sample's distance from this threshold, prioritizing high-confidence examples. Experiments on visual generative models show UPO improves over a supervised fine-tuning baseline.

**Strengths:**

1.Important Problem: The paper tackles a significant and practical limitation of DPO. Paired preference data is costly to collect, while unpaired scalar scores are a far more common form of real-world feedback. A stable, effective method for this data modality would be highly valuable.

2.Simplicity and Novelty: The proposed method is simple, intuitive, and easy to implement. Using a global threshold to generate "pseudo-labels" is a novel adaptation of the preference optimization framework that cleverly sidesteps the need for explicit pairs.

**Weaknesses:**

1. Fundamentally Flawed Core Assumption: The central thesis of UPO is replacing the intractable, instance-dependent oracle baseline $\tau^*(x) = \beta \log Z(x)$ with a global, static threshold $\tau$. This is a major oversimplification. The reward distribution for a prompt $x$ is highly dependent on the prompt's complexity. A "good" image for a very difficult prompt may receive a lower absolute score than a "mediocre" image for a very simple prompt. UPO's global threshold will incorrectly label the good, hard-prompt image as "pseudo-rejected" and the mediocre, easy-prompt image as "pseudo-preferred," actively steering the model in the wrong direction.

2. Weak Experimental Baselines: The primary comparison in the paper is against "Supervised Fine-Tuning (SFT)," which is defined as fine-tuning only on samples with scores above the threshold $\tau$. This is a known weak baseline that often suffers from mode collapse and overfitting to the high-reward subset. The paper fails to compare UPO against more relevant and robust baselines, such as:
  - Standard RLHF (e.g., PPO) using the scalar scores directly as rewards.
  - A DPO variant where pairs are synthesized from the scalar data (e.g., by sampling two images for the same prompt and using their scores to create a $(y_w, y_l)$ pair).
  - Other methods from the literature designed for scalar rewards (like AWR or the cited QRPO). This lack of strong comparison makes the empirical results unconvincing.

3. Mismatch Between Theory and Practice: The theoretical analysis (Theorem 2.1, Appendix B) provides standard consistency proofs, showing that the UPO estimator converges to the optimum of the UPO loss function. However, this analysis does not prove that optimizing this proxy loss (with its flawed global threshold) actually optimizes the true, original KL-regularized alignment objective. There is a significant gap between what the theory validates (convergence of the proxy) and what it needs to validate (that the proxy is a faithful and effective approximation).

4. Poor Online Performance: The "Online UPO" variant (Appendix D, Table 3), which should be more adaptive, performs worse than the offline version. This suggests the global thresholding mechanism is highly unstable and sensitive to the data used for its estimation, undermining its potential for practical, large-scale deployment.

**Questions:**

1. The entire method hinges on the global threshold $\tau$ being a meaningful proxy for the instance-dependent baseline $\tau^*(x)$. How can this hold when reward distributions vary so much by prompt? Have the authors analyzed the failure cases this creates, such as when high-quality images for hard prompts are incorrectly penalized?

2. Why was SFT (fine-tuning on the top 50% of data) chosen as the main baseline? This seems like a weak competitor. Why not compare against a standard RL algorithm (like PPO or AWR) that uses the scalar scores directly as a reward signal?

3. The related work mentions QRPO (Matrenok et al., 2025), which also aligns models from scalar rewards by solving the partition function in a different, more principled way (quantile transformation). This seems to be a direct and superior competitor. Was a comparison considered?

4. Instead of a single global threshold, would a more reasonable approximation be a prompt-conditioned threshold, $\tau(x)$, perhaps learned by a simple predictive model? This seems like a more robust way to approximate the instance-dependent baseline.

5. The confidence weighting $w(s,\tau)=1+c\cdot|s-\tau|$ gives more weight to samples far from the median. Could this cause the model to overfit on "easy" cases (very good or very bad images) and ignore the more nuanced samples near the decision boundary, which are arguably the most important for fine-grained alignment?

---

### Official Review · Reviewer_RLPm · 2025-10-31

**Soundness:** 3
**Presentation:** 3
**Contribution:** 3
**Rating:** 4
**Confidence:** 4

**Summary:**

This paper introduces Unpaired Preference Optimization (UPO), a new framework to align generative models using unpaired reward scores, addressing the limitation of DPO which requires paired preference data and scores.
The core idea is to use a global threshold (like the median score) to classify samples as "pseudo-preferred" or "pseudo-rejected."  A confidence-weighting mechanism gives more importance to samples with scores far from the threshold, fully leveraging the score's magnitude.
Theoretically, this approach is justified as a tractable proxy for the ideal but intractable KL-optimal decision rule, which compares a sample's reward to an oracle baseline. Experiments on visual models show UPO outperforms a standard supervised fine-tuning (SFT) baseline.

**Strengths:**

UPO's design is backed by a rigorous theoretical derivation. It elegantly reframes the intractable KL-regularized RL objective, which compares a sample's reward to an ideal baseline, into a simple and tractable classification task using a global threshold. The derivation is clear and persuasive, resulting in a final loss function that is easy to implement.

**Weaknesses:**

1. **Insufficient Motivation**: The paper's core motivation—unpaired data is difficult to use for preference optimization—is not entirely convincing. It is straightforward to construct preference pairs from reward scores (e.g., by pairing the highest- and lowest-scoring samples within a batch), which undermines the stated necessity for a new framework like UPO.
2. **Weak Experimental Comparison**: The empirical evaluation is insufficient in two key aspects:
  - **Missing Baselines**: The paper fails to compare against strong and relevant baselines. Crucially, it omits simple yet effective strategies like applying DPO on pairs constructed from scalar scores. It also neglects to compare against recent state-of-the-art methods designed for similar settings, such as Flow-DPO [1] and Flow-GRPO [2]. This absence makes it impossible to judge whether UPO offers a tangible advantage.
  -  **Limited Scope**: The experiments are conducted on relatively older models. There are no results on more advanced and powerful models like the diffusion-based FLUX [3] and Qwen-Image [4], or the autoregressive NextStep-1 [5]. Furthermore, evaluation is not performed on established, general benchmarks like GenEval[6], which limits the generalizability of the findings.
3. **Potential Limitation of the Global Threshold**: The use of a single global threshold is a major simplification that may not be robust. It fails to account for variations in prompt difficulty, where a high-quality output for a challenging prompt might receive a lower absolute score than a mediocre output for a simple prompt. This could lead to systematic mislabeling of pseudo-preferences, an issue the paper does not address.
4. **Limited Theoretical Novelty**: While the theoretical derivation is sound, its primary contribution lies in the elegant re-formulation of the alignment problem. The underlying proof techniques rely on standard statistical methods and do not introduce fundamentally new theoretical tools.

[1] Diffusion model alignment using direct preference optimization

[2] Flow-GRPO: Training Flow Matching Models via Online RL

[3] Black Forest Labs. Flux. https://github.com/black-forest-labs/flux, 2024.

[4] Qwen-Image Technical Report

[5] NextStep-1: Toward Autoregressive Image Generation with Continuous Tokens at Scale

[6] GenEval: An Object-Focused Framework for Evaluating Text-to-Image Alignment

**Questions:**

1. **Motivation & Baseline Justification**: Could you clarify the specific advantages of UPO over simpler DPO-based baselines where pairs are constructed by comparing reward scores? To validate UPO's effectiveness, comparisons against these simple baselines, as well as recent methods like Flow-DPO [1] and Flow-GRPO [2], are essential.
2. **Broader Empirical Validation**: To demonstrate the robustness and scalability of UPO, could you provide results on more advanced models (e.g., FLUX [3], Qwen-Image [4], NextStep-1 [5]) and standard, holistic benchmarks like GenEval [6]?
3. **Thresholding Robustness**: How does UPO handle "hard prompts" whose best outputs may consistently fall below the global threshold? Furthermore, an ablation study with a fixed or random threshold would be valuable to quantify the benefit of your dynamic, percentile-based thresholding mechanism.
4. **Broader Empirical Validation**: To demonstrate the robustness and scalability of UPO, could you provide results on more advanced models (e.g., FLUX [3], Qwen-Image [4], NextStep-1 [5]) and standard benchmarks like GenEval [6]?
5. **Thresholding Robustness**: How does UPO handle "hard prompts" whose best outputs may consistently fall below the global threshold? Additionally, an ablation study with a fixed or random threshold would help quantify the benefit of your dynamic thresholding mechanism.

---

### Note · Authors · 2025-11-12

I have read and agree with the venue's withdrawal policy on behalf of myself and my co-authors.